# Cutaneous and Pulmonary Tuberculosis—Diagnostic and Therapeutic Difficulties in a Patient with Autoimmunity

**DOI:** 10.3390/pathogens12020331

**Published:** 2023-02-15

**Authors:** Monika Kozińska, Ewa Augustynowicz-Kopeć, Andrzej Gamian, Anna Chudzik, Mariola Paściak, Przemysław Zdziarski

**Affiliations:** 1Department of Microbiology, National Tuberculosis and Lung Diseases Research Institute, Plocka 26, 01-138 Warsaw, Poland; 2Department of Immunology of Infectious Diseases, Hirszfeld Institute of Immunology and Experimental Therapy, Polish Academy of Sciences, Rudolfa Weigla 12, 53-114 Wroclaw, Poland; 3Department of Clinical Immunology and Pulmonary Disease, Lower Silesian Oncology, Pulmonology and Hematology Centre, P.O. Box 1818, 50-385 Wroclaw, Poland

**Keywords:** cutaneous tuberculosis, extrapulmonary TB, pulmonary TB, *Mycobacterium tuberculosis* complex, skin infections

## Abstract

Cutaneous tuberculosis (CTB) is a very rare disease and accounts for only 1–2% of cases of extrapulmonary tuberculosis (EPTB). Due to the variety of its clinical manifestations, the uncharacteristic appearance of its lesions, resembling other dermatoses in the early stages, and the limited experience of clinicians due to the rarity of CTB, diagnosis is very difficult. Particularly noteworthy is the fact that most cases of EPTB, including skin tuberculosis (TB), can be a manifestation of systemic involvement. In this paper, we present a case of an immunocompromised patient who was diagnosed with CTB almost a year after the first dermatological lesions were located on the lower extremities. At the same time, due to respiratory symptoms, a diagnosis of pulmonary TB (PTB) was made, and radiological and microbiological confirmations were obtained.

## 1. Introduction

Tuberculosis (TB) is a systemic infection caused by *Mycobacterium tuberculosis* complex (MTBC). It is droplet-transmitted and most commonly manifests itself in the respiratory tract (pulmonary TB–PTB). However, it can cause disease in any system or organ (extrapulmonary TB–EPTB) [1]. The dissemination of the disease from the primary infection site in the lungs can occur at the time of primary infection, or many years later when immune control fails. EPTB remains a critical concern both in developing and developed countries, accounting for 15% to 30% of all tuberculosis cases [2]. In persons with HIV/AIDS and other immunocompromised states, it is a prevalent opportunistic infection. The EPTB form most commonly involves the lymph nodes (19%), pleura (7%), gastrointestinal tract (4%), bones (6%), central nervous system (3%), genitourinary system (1%), and other organs [3]. A diagnostically, clinically, and therapeutically special population of TB patients are those with coexisting PTB and EPTB forms. EPTB without lung involvement accounts for an average of 19% of all new TB cases, ranging from 6% to 44%, while EPTB with lung involvement accounts for approximately 6% of cases [4].

In Poland, EPTB is rarely recorded, accounting for only 4.6% of all TB cases, and most commonly involves pleural TB [5].

Cutaneous TB (CTB) is a very rare disease comprising only 1–2% of EPTB cases [6,7]. CTB more frequently affects women, especially young adults, and its lesions are mainly located on the face, neck, and abdomen, or in other sites that have been in contact with mycobacteria [7,8].

Due to the variety of its clinical manifestations and the non-specific occurrence of its lesions that resemble other types of skin disease at an early stage, the diagnosis of CTB is very difficult. In addition, the limited experience of clinicians, due to the rarity of CTB, means that they often do not suspect a tuberculous aetiology and do not diagnose the patient in this direction [6]. Importantly, in most cases, EPTB, including CTB, may be a manifestation of systemic involvement. Obviously, patients diagnosed with CTB should also be tested for the potential tuberculosis of other organs and systems. The search for and collection of material from all lesions in a patient with PTB, to assess the actual spread of the disease, is a less obvious clinical practice.

In this article, we describe the case of a patient with microbiologically confirmed CTB and lesions located on the lower limb. At the initial stage of infection, the lesions were diagnosed as the result of rheumatoid nodules; thus, the patient was treated with topical steroid therapy (ineffective).

During the months-long diagnostic process, PTB and CTB were diagnosed. *Mycobacterium tuberculosis* complex strains, cultured from both sites, presented an identical phenotype and genotype, suggesting that the PTB was most likely the primary disease and source of the spreading infection.

## 2. Case Study

A 64-year old non-smoking man (weight, 59 kg; height, 180 cm; and BMI value, 18 kg/m^2^) of Caucasian ethnicity, with a high fever, dyspnoea, and acute respiratory distress syndrome, was admitted to the Lower Silesian Oncology, Pulmonology and Hematology Centre after computer tomography (CT) was performed at the one of the Polish rheumatology hospitals where he had been treated for years. A CT scan (unavailable) showed pleural thickening of the apexes of both lungs, with small thick-walled cavities. Small cavities (the largest was 29 mm) appeared at the apex of the left lung.

His medical history included ankylosing spondylitis (AS), rheumatoid arthritis (RA), and ventilation disorders caused by a significant deformation of the chest. Because of evidence of spinal instability (vertebral fracture) at the age of 14, he was treated with a spinal column stabiliser (Figure 1).

X-ray studies were repeated every year, mainly for a bone examination of his RA or post-medical osteoporosis (with the rheumatologic indication). The patient’s performance was constantly deteriorating. The X-ray alterations of an upper lobe of the lung were observed and interpreted by rheumatologist as a systemic manifestation of AS and nontuberculous fibrosis (probably an adverse drug reaction to the disease-modifying treatment of RA and AS).

Previously (about 10 months ago), skin lesions (nodules and ulcers) were identified in various locations, primarily on the limbs. Topical therapy with steroids was ineffective, and the lesions were diagnosed as the results of rheumatoid nodules. Fatigue, anorexia, and weight loss were observed later. The patient was referred to the rheumatologic hospital by a rheumatologist because of increasing bone pain and dyspnoea, as well as hypergammaglobulinaemia, proteinuria, and high leukocytosis in laboratory tests. After haematological examinations, he was diagnosed with probable plasmocytic dyscrasia (although without monoclonal antibody). As emergency medications, the patient received aggressive glucocorticoids and cyclophosphamide. CT was then performed for the dyspnoea (see above), and the presence of cavities led to the suspicion of tuberculosis. The patient was transferred to the Lower Silesian Oncology, Pulmonology and Hematology Centre under the care of a specialist in infectious diseases and clinical immunology.

Upon physical examination in our centre, we observed local lymphangitis and oedema of the extremities with atypical lesions (single, painless, non-bleeding skin ulcers) (Figure 2A,B).

Bronchoalveolar lavage (BAL) was performed in May 2021 to collect a sample of material for microbiological testing of the lesions in the lungs. The clinical specimens were cultured on solid Löwenstein–Jensen (L–J) and liquid growth media in the Bactec MGIT 960 system (Becton Dickinson Diagnostic Systems, Sparks, MD, USA). After 9 days of incubation on a liquid medium and after 3 weeks on the L–J medium, the growth mycobacteria’s susceptibility to first-line TB drugs was observed. The critical concentrations for the specific drugs were as follows: isoniazid (INH), 0.2 μg/mL; rifampicin (RIF), 40 μg/mL; ethambutol (EMB), 2 μg/mL; streptomycin (STR), 4 μg/mL (for the L–J medium); and INH, 0.10 μg/mL; RIF, 1.00 μg/mL; EMB, 5.00 μg/mL; STR, 1.00 μg/mL; and pyrazinamide (PZA), 100 μg/mL (for the liquid medium). During that period, swabs from the wounds located on the lower extremities were also collected for microbiological tests (Figure 2).

After several weeks, *Mycobacterium tuberculosis* strains, susceptible to first-line TB drugs, were cultured.

The strains cultured from the BAL samples and wound swabs were sent to the National Reference Laboratory for Tuberculosis (NRLT) for genotyping. The genetic relationship between the isolated strains was investigated by spoligotyping (Ocimum Biosolutions, Hyderabad, India) and 15-*loci* MIRU-VNTR (Mycobacterial Interspersed Repetitive Units-Variable Number of Tandem Repeat) analysis [9,10]. A 100% homology of the isolates was confirmed by both genotyping methods—both strains presented an ST 53 spoligotype and 333633242232225 MIRU-VNTR DNA profile. The patient was treated according to the following antimycobacterial regimen: RMP, INH, PZA, and EMB.

The clinical sequelae included anaemia, hypoalbuminaemia, and paraproteinaemia with renal symptoms, a weight loss of 5 kg, and finally multiple organ dysfunction syndrome. The patient died 2 months later.

## 3. Discussion

CTB is a special form of TB due to its very diverse clinical presentation and the mild and chronic course of the disease, resulting from the unfavourable conditions for the growth of mycobacteria in the skin, and other factors [11]. The lesions caused by the mycobacterium species vary from small papules (e.g., primary inoculation tuberculosis) and warty lesions (e.g., tuberculosis verrucosa cutis), to massive ulcers (e.g., Buruli ulcers) and plaques (e.g., lupus vulgaris) that can be highly deformative [6]. Most frequently, cutaneous leishmaniosis, leprosy, infections with atypical mycobacteria, syphilis, actinomycosis, fungal infections and sarcoidosis, and other diseases are considered in the differential diagnosis of CTB [12]. Due to the various modes of clinical manifestation, the uncharacteristic appearance of the lesions, resembling other dermatoses in the early stages of the disease, and the limited experience of clinicians due to the rarity of CTB, correct diagnosis is very difficult.

Therefore, when diagnosing CTB, special attention should be paid not only to the phenotype of the lesions, but also to all the factors predisposing the patient to infection. Otherwise, it is easy to overlook the actual cause of the disease.

Apart from the difficulties associated with a wide range of differential diagnoses, there are significant problems with the microbiological confirmation of clinically suspected CTB. Therefore, in addition to standard microbiological procedures, it is mandatory to perform molecular tests that offer high sensitivity and specificity [13]. The importance of histopathological analysis (neglected in the presented case) as a diagnostic tool in CTB should be emphasised here, especially if microbiological tests are negative [14,15]. The described clinical case draws attention to a very important clinical and epidemiological problem, which is the coexistence of PTB and CTB, without the signs of miliary TB and the involvement of other organs. CTB is rarely the primary form of MTBC infection, and is usually a consequence of infection caused by primary PTB, or a manifestation of systemic involvement. Due to this fact, the diagnosis of a patient with suspected CTB should also consider pulmonary involvement [16,17,18,19].

To summarise, the presented case of a patient with CTB and PTB is an example of the systemic comorbidities (onco-haematological, rheumatoid) associated with immunodeficiency (also post-therapeutic). In such cases, the diagnostic procedures should focus on opportunistic infections and latent infection reactivation, including TB, which may have a different manifestation in each organ because of the primary disease. The list of skin conditions is open in this context, and even allergy-specific lesions may be associated with mycobacterial infection [20].

## Figures and Tables

**Figure 1 pathogens-12-00331-f001:**
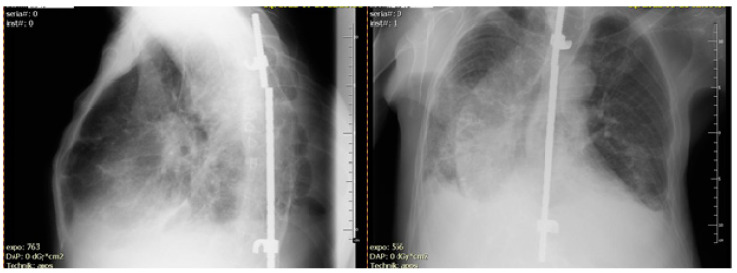
X-ray examination during admission to the Department of Pulmonary Diseases. Overlapping post-inflammatory fibrosis and cavities were observed.

**Figure 2 pathogens-12-00331-f002:**
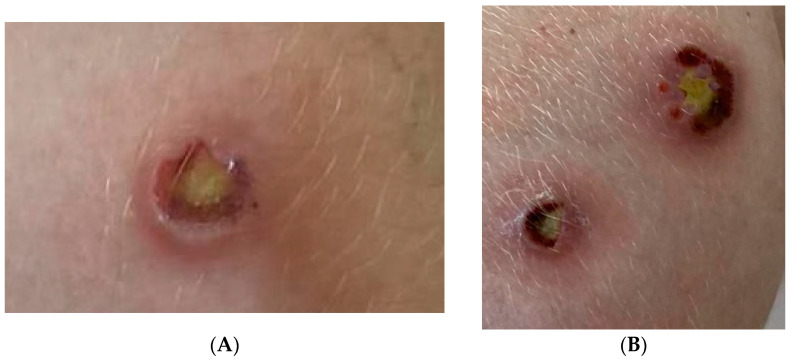
Single, painless, non-bleeding skin ulcers located on the lower extremities (**A**,**B**).

## Data Availability

All data generated or analysed during this study are included in this article. The mycobacterial strains were deposited in the National Tuberculosis and Lung Diseases Research Institute, Warsaw, Poland.

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
