# Peer review of "Cutaneous and Pulmonary Tuberculosis—Diagnostic and Therapeutic Difficulties in a Patient with Autoimmunity"

_pathogens, 2023, doi:10.3390/pathogens12020331_

Round 1

Reviewer 1 Report

I read with interest a case by Kozińska et al. about a comorbid man with concomitant cutaneous and pulmonary tuberculosis (TB). The case seems to describe the difficulties about diagnosing skin TB and underlying pulmonary TB. The case is interesting and there are many good learning points from the case although I think the case needs some improvement. The language could be improved considerably, specifically sentence syntax, grammar, and figure legends, but also the timeline and structure. Some parts of the case are also poorly described. Lastly, I think the aim of the case and learning points are a little unclear throughout the manuscript. My specific comments are provided below. Consequently, I suggest major revision.

Abstract
I think the sentence line 21-24 is a little clumsy and could be paraphrase. The content is fine. You could include the specific time (e.g. days or months) from first symptoms to diagnosis and to treatment to underline the delay and your aim with the case. If so, this should also be implemented in the main text.

Why was genotyping done to confirm that skin and pulmonary TB was due to the same strain? That seems unnecessary if the bacteria are found in both localization at the same point in time? What is known about dual infection with different Mtb strains?

Introduction

Lines 35-39, also a little bit long sentence that could be changed or split in two.

You could add, how many percentages of EPTB and CTB that are typically considered to have concomitant systemic and/or pulmonary involvement?

Lines 44-48, your statement does not really justify why you wrote the case. CTB is not a new manifestation – with or without PTB – and you don’t really explain what the particular interest of this case is here.

Case presentation

Line 50, what do you mean with systemic inflammatory disease? Please be more specific or state symptoms leading (skin lesion, fever of unknown origin, weight loss, etc) to health care contact?

Line 52-54, again, please be specific. I suggest paraphrasing to something like: “The patient had a medical history of ankylosing spondylitis (AS), rheumatoid arthritis (RA) and….”.
“Rheumatic diseases” does not give the reader any additional information. Unless the patient had additional rheumatic diseases? Also, please be specific regarding ventilation disorder… restrictive lung disease due to abnormal anatomy (or
kyphosis, pectus carinatum, pectus excavatum or what else you may be referring to)

Line 56, X-ray studies were repeated every year mainly for bone examination? Due to his SpA?

Line 56-57, “The patient's performance was constantly deteriorating.” Due to…? Bone instability? Lung function? Fever, weight loss?

Line 56-58, “Systemic manifestation of AS and nontuberculous fibrosis of an upper lobe of the lung was observed.” ILD due to SpA and/or RA? The sentence is unclear.

Figure 1: There is lacking an ”s” at the end of “Department of Pulmonary Disease”. The text needs to be improved. I am not sure of the message here in the text.

Line 64, previous to the presentation with systemic symptoms? How much later did fatigue, anorexia and weight loss start?

Line 65-66, rheumatoid nodules are usually at first treated with injections of glucocorticoid. That does not seem to be the case here even though you suspected that? If you have clinical pictures of the skin lesions and the nodules at first, they could be presented together with figure 2. If these are not the same lesions?

Line 68-69, “The patient was referred to the hospital by a rheumatologist because of the increasing bone pain, dyspnoea”. But in line 50, you state he was admitted to the hospital? Admitted to another hospital?

Figure 2: “…observed by patient in initial phorm.”I don’t understand

Line 72, “….hematological examinations” = bone marrow examination? What plasmocytic dyscrasia was suspected?

Line 74, You could provide CT scans instead, or together with the xray, to show these as the xray is a little bit poor quality. Then cavities, fibrosis etc. would be more easy to show.

Line 77, please remove specific date to protect the identity of the patient. Better with X months or just “May 2021”.

You could consider limiting the mentioning of microbiological methods a little bit as it is of little relevance to the case. Also, in line 85-86, you could write “… the growth mycobacteria suscpetible to first line TB drugs”.

Line 91, why abbreviated to AFB- as done in line 79 if you don’t use the abbreviation?

Was material not sent for pathohistological examination?

Did the patient have any history of TB or exposure to Mtb, or other risk factors for TB, besides immunosuppression?

You could consider stating in a brief sentence in the beginning of the section, in what country the patient was treated as this tells us something about the underlying risk of Mtb and prevalence of HIV for instance.

Discussion

I think the first part of the discussion is very fine. However, I think a major issue, and cause of the delay in diagnosis, is not mainly the challenging diagnostics – at least not if decent labs are available – but that clinicians may be slow to consider the diagnosis if not used to working with TB. You could also discuss or briefly mention some of the different clinical manifestations of skin TB if the case is to provide some learning for the reader.

Line 136, what do you mean with iatrogenic? Unclear to me

Line 137, tuberculosis is not the infection (=MTBC) but the disease.

Reviewer 2 Report

This is an interesting rare case report of high clinical relevance. 

Minor comments

1. The authors could add basic physical exam details such as Height, weight, BMI and others

2. Was there any pulmonary symptoms in the patient such as persistent cough ? 

3. The authors must add details on what was the drug concentration followed in MIC assay in both MGIT and LJ for INH, SM, RIF, EMB and PZA

Author Response

The authors thank for the review, all valuable comments and suggestions from the Reviewer. We have made a great many changes to the manuscript and hope that it has become more valuable and educational.

Comments and Suggestions for Authors

This is an interesting rare case report of high clinical relevance. 

Minor comments

  1. The authors could add basic physical exam details such as Height, weight, BMI and others

Answer

As suggested, data have been inserted in the text.

  1. Was there any pulmonary symptoms in the patient such as persistent cough ? 

Answer

Yes, dyspnoea and dry cough. These data were added to the text.

  1. The authors must add details on what was the drug concentration followed in MIC assay in both MGIT and LJ for INH, SM, RIF, EMB and PZA

Answer

As suggested, critical critical concentrations for drugs were introduced in the manuscript.

Reviewer 3 Report

This case report is of clear clinical interest, especially in countries where tuberculosis is not an endemic disease. It is well presented with a wealth of details, which makes it very didactic.

I suggest that authors pay attention in the Discussion to the chest X-ray alterations described as ....."Systemic manifestation of AS and nontuberculous fibrosis of an upper lobe of the lung was observed." In fact, this non TB fibrosis is a it is a lesion related to previous TB (old lesions) common in adults who had contact with a source case at some point in their lives. I suggest commenting on this  because this finding would be a diagnostic clue.

Author Response

The authors thank for the review, all valuable comments and suggestions from the Reviewer. We have made a great many changes to the manuscript and hope that it has become more valuable and educational.

Comments and Suggestions for Authors

This case report is of clear clinical interest, especially in countries where tuberculosis is not an endemic disease. It is well presented with a wealth of details, which makes it very didactic.

I suggest that authors pay attention in the Discussion to the chest X-ray alterations described as ....."Systemic manifestation of AS and nontuberculous fibrosis of an upper lobe of the lung was observed." In fact, this non TB fibrosis is a it is a lesion related to previous TB (old lesions) common in adults who had contact with a source case at some point in their lives. I suggest commenting on this  because this finding would be a diagnostic clue.

Answer

Indeed the patient was born and grew up in difficult times in post-war Poland, where the detection and curability of tuberculosis was low. Fibrous changes in X-rays (observed by rheumatologists), in turn, are often associated with side effects of RA modifying drugs. Because of late TB diagnosis patient did not receive more immunosuppressive therapy and AS/RA progression with fatal sequel was cause of death. The tuberculosis was stopped.

Thank you for the suggestion.

Round 2

Reviewer 1 Report

Thank you for your interesting case and reponse to my comments

Author Response

Dear Reviewer,

Thank you for your comments and suggestions. We have once again done a language proofread and corrected the errors.
We have left the results in their previous form, we hope you will accept the version.
